# Nitrogen Fertilization Modified the Responses of *Schima superba* Seedlings to Elevated CO_2_ in Subtropical China

**DOI:** 10.3390/plants10020383

**Published:** 2021-02-17

**Authors:** Zhan Chen, Siyuan Ye, Jixin Cao, He Shang

**Affiliations:** Key Laboratory of Forest Ecology and Environment of National Forestry and Grassland Administration, Research Institute of Forest Ecology, Environment and Protection, Chinese Academy of Forestry, Beijing 100091, China; chenzhan0508@caf.ac.cn (Z.C.); ysy1994@126.com (S.Y.); caojix@163.com (J.C.)

**Keywords:** elevated CO_2_, N fertilization, leaf N, photosynthesis, biomass

## Abstract

There are very few studies about the effects of relatively higher CO_2_ concentration (e.g., 1000 μmol·mol^−1^) or plus N fertilization on woody plants. In this study, *Schima superba* seedings were exposed to ambient or eCO_2_ (550, 750, and 1000 μmol·mol^−1^) and N fertilization (0 and 10 g·m^−2^·yr^−1^, hereafter: low N, high N, respectively) for one growth season to explore the potential responses in a subtropical site with low soil N availability. N fertilization strongly increased leaf mass-based N by 118.38%, 116.68%, 106.78%, and 138.95%, respectively, in different CO_2_ treatments and decreased starch, with a half reduction in leaf C:N ratio. Leaf N was significantly decreased by eCO_2_ in both low N and high N treatments, and N fertilization stimulated the decrease of leaf N and mitigated the increase of leaf C:N by eCO_2_. In low N treatments, photosynthetic rate (Pn) was maximized at 733 μmol·mol^−1^ CO_2_ in August and September, while, in high N treatments, Pn was continuously increased with elevation of CO_2_. N fertilization significantly increased plant biomass especially at highly elevated CO_2_, although no response of biomass to eCO_2_ alone. These findings indicated that N fertilization would modify the response of *S. superba* to eCO_2_.

## 1. Introduction

Elevated atmospheric CO_2_ concentration (eCO_2_) and N deposition are always important issues in the research of global change. Until today, amounts of previous studies reported the effects of eCO_2_ alone or eCO_2_ combined with N deposition on plants [1,2,3,4,5,6]. While, according to the IPCC (Intergovernmental Panel on Climate Change) Fifth Evaluation Report, the CO_2_ concentrations will be between 430–480, 580–720, 720–1000 μmol·mol^−1^, and more than 1000 μmol·mol^−1^ in 2100 without additional mitigation efforts, respectively, in the four levels of Representative Concentration Pathways (RCPs), RCP2.6, RCP4.5, RCP6.0, and RCP8.5 [7]. RCPs are used for making projections based on these factors, described four different 21st century pathways of greenhouse gas emissions and atmospheric concentrations, air pollutant emissions, and land use in this report [7]. N_2_O emission will also increase with increasing CO_2_ concentration, and will reach up to more than 20 T g·yr^−1^ until 2100 for scenario RCP8.5 [7]. And total reactive-N emissions were predicted to reach 83 and 114 Tg N yr^−1^ by 2100 under the RCP4.5 and RCP8.5, respectively [8]. A model analysis over the globe indicated that large increases have occurred for all soluble reactive N fractions and will continuously occur mainly in Asia in the future, and NO_y_ deposition will be increased only in Asia [9]. Across all forms of deposition, rates of N deposition in Eastern Asia are among the highest in the world, in which China is a hotspot of N deposition with the greatest number of urban sites mentioned in 174 publications focusing on N deposition over the world [10].

Most of the previous experiments studied the effects of double ambient CO_2_ concentration (about 700 μmol·mol^−1^) on plants, which could not predict the potential responses of the plants to the future CO_2_ concentration based on RCP6.0 and RCP8.5. Although a few reports studied the effects of the higher CO_2_ concentration like 1000 μmol·mol^−1^ on the crops [11,12,13,14,15,16,17], to the best of our knowledge, there have been no studies on the effects of about 1000 μmol·mol^−1^ CO_2_ combined with N fertilization on woody plants.

This study aimed to explore the responses of native tree species to highly elevated CO_2_ and N fertilization. *S. superba* is a representative tree species of subtropical forests in China and widely used in urban landscaping. Previous studies showed that photosynthesis, N:P ratio, biomass accumulation and allocation, and soil respiration of *S. superba* were affected by eCO_2_ (about 700 μmol·mol^−1^) and N deposition [18,19,20]. Photosynthetic rate of *S. superba* seedlings was reduced, while height growth was enhanced by eCO_2_, after the exposure of 6 months [20]. Although eCO_2_ did not affect photosynthetic rate in high N treatment, *S. superba* grown in ambient N treatment had 23% and 47% greater photosynthate rate (Pn) in ambient CO_2_ than those in eCO_2_ after exposure of 20 and 31 months, and leaf N was also reduced by eCO_2_ in both low and high N treatment [21]. If this species similarly responds to high CO_2_ concentration as to about 700 μmol·mol^−1^ CO_2_ in previous studies, how would it respond to high CO_2_ and N fertilization? In this present study, we tried to explore the responses of gas exchange, carbohydrates and biomass to different levels of CO_2_ concentration and N fertilization. We hypothesized that: 1) Pn, leaf N would be reduced much more at 1000 μmol·mol^−1^ CO_2_ than at 700 μmol·mol^−1^ CO_2_; and 2) nitrogen fertilization would affect the response of *S. superba* to eCO_2_.

## 2. Results

### 2.1. Leaf N, C:N Ratio, and Carbohydrates

Leaf N was decreased with CO_2_ concentration increasing at both low N and high N treatments (Figure 1, R^2^ = 0.9968, *p* = 0.003 and R^2^ = 0.8798, *p* = 0.034, respectively). High N significantly improved leaf N (*p* ≤ 0.05) (Figure 1) by 118.38%, 116.68%, 106.78%, and 138.95% compared to low N, respectively, at ambient air, 550, 750, and 1000 μmol·mol^−1^ CO^2^. Although soluble sugar content was not significantly affected by elevated CO_2_ at low N, while it was linearly decreased with CO_2_ enrich at high N (Figure 1, R^2^ = 0.8580, *p* = 0.016). The starch content was linearly increased by eCO_2_ under both low N and high N, while the increment at high N was slower than that at low N (Figure 1). N fertilization decreased the starch contents by 31.52%, 49.36%, 39.32%, and 52.00%, respectively, at ambient, 550, 750, and 1000 μmol·mol^−1^ CO_2_. Simultaneously, the accumulation of leaf N and reduction of sugar and starch by high N leaded to reduction in plant C:N ratio by 51.64%, 54.79%, 50.92%, and 59.67%, respectively, at ambient and elevated CO_2_ concentration under N fertilization (Figure 1). Although leaf C:N was both increased with CO_2_ concentration increasing at low and high N, the slope of the regression equation at high N was lower than that at low N.

### 2.2. Gas Exchange

In general, Pn, and g_s_ both decreased from July to September (Figure 2 and Figure 3). In July, Pn was increased with CO_2_ enrich at both low N and high N, with greater slope at high N (Figure 2). Pn was also linearly increased by eCO_2_ at high N in August and September (*p* < 0.001), while, at low N, Pn was increased to the maximum at 733 μmol·mol^−1^ CO_2_ and then decreased when CO_2_ concentration continuously increased (Figure 2). Pn from plants exposed to 1000 μmol·mol^−1^ CO_2_ at high N was increased of 73.68%, 119.90%, and 183.46%, respectively, measured in July, August, and September compared to ambient CO_2_ at low N. CO_2_ concentration, N fertilization and measuring time all had significant effects on Pn (*p* < 0.001), and there were significant interactions between CO_2_ and N, CO_2_ and measuring time, N and measuring time, while no significant impacts of CO_2_, N and measuring time on Pn. No matter under low N or high N, g_s_ was significantly negatively correlated with eCO_2_ in all measuring time (*p* < 0.001 at both low N and high N in July and August; *p* = 0.009 and 0.015, respectively, at low N and high N in September. In July, the two lines were almost parallel at both low N and high N with almost equal slopes (0.02259 and 0.02300, respectively, at low N and high N), while, in August and September, g_s_ decreased more quickly with CO_2_ concentration increasing at high N than that at low N, with steeper lines at high N. Whether it is CO_2_ concentration, N fertilization, or measuring time, all had significant impacts on g_s_, while there was no interaction among them (Figure 3).

### 2.3. Plant Biomass

Elevated CO_2_ alone even up to 1000 μmol·mol^−1^ had no effects on plant biomass, including different issue and total biomass, whether at low N or high N. While N fertilization impacted the biomass at eCO_2_. In ambient CO_2_ treatments, N fertilization had no effect on biomass. High N significantly increased leaf biomass and total biomass compared to plants in low N treatments at 550 μmol·mol^−1^ CO_2_. When plants exposed to 750 μmol·mol^−1^, N fertilization increased root and total biomass. Notably, N fertilization significantly increased the biomass of each issue and also the total biomass when plants exposed to 1000 μmol·mol^−1^. The results of two-way ANOVA indicated that eCO_2_ had not any significant effects on plant biomass, while N fertilization significantly affected biomass (*p* < 0.001), without interaction between CO_2_ and N fertilization (Table 1).

## 3. Discussion

Many studies have shown that eCO_2_ caused a reduced N concentration, a decrease in g_s_ and an increase of starch accumulation [22,23], all of which were all confirmed by this present study. In this study, eCO_2_ had a significant negative effect on leaf N in both low and high N treatments, and high N markedly increased leaf N compared to low N in all CO_2_ treatments, which partially confirmed our first hypothesis that the reduction of leaf N would be much greater at 1000 μmol·mol^−1^ CO_2_ than at 700 μmol·mol^−1^ CO_2_. The reduction rate of leaf N induced by eCO_2_ under high N treatments was quicker and the slope was larger than that under low N treatments (Figure 1). The possible reasons for leaf N decline in low N treatments include less N available [24] and CO_2_ inhibition of nitrate assimilation [25]. While the rapid reduction of leaf N by eCO_2_ in high N treatments could be interpreted as a dilution effect [26,27], because high N highly stimulated plant biomass (Figure 4). The study conducted in tropical China showed that eCO_2_ increased leaf N of *S. superba*, while N fertilization had no effect leaf N [28], which was contrasted with our results that N fertilization affected the response of leaf N to eCO_2_. The different responses of *S. superba* to eCO_2_ and N fertilization between the two studies were attributed to N availability. At tropical site in China, N was not a limiting factor due to high ambient N deposition [28], while, in our experimental site, plants were constrained by N and P [29].

The reduction of leaf N caused by eCO_2_ led to imbalance of C and N, and the C:N ratio increased with the increase of CO_2_ concentration. While, it is pivotal to maintain the C:N ratio for various growth and development processes in plants productivity [22]. N fertilization alleviated the effects of eCO_2_ on imbalance of C and N. Although eCO_2_ could still increase leaf C:N under high N treatments, the slope of linear regression across CO_2_ concentration was much smaller than that under low N treatments (Figure 1). The increment of C:N ratio could increase a large accumulation of carbohydrates [30,31,32] that cannot be used for structural growth, which was also confirmed in this present study that starch content was significantly accumulated under higher CO_2_ concentration in both low N and high N treatments, and the slope of high N treatment was much lower than that of low N (Figure 1). Conversely, lower starch concentrations in high N seedlings than in low N seedlings imply that N fertilization mitigate the inhibition of growth by eCO_2_ because starch accumulation is a passive response to decreased rates of growth [33]. This was confirmed by the plant biomass in our study, which was not affected by eCO_2_, and the plant biomass was greatly increased when N was applied at high CO_2_ concentration (Figure 4).

Although there are exceptions to the general rule that g_s_ declines under eCO_2_, in Free Air CO_2_ Enrichment experiments, g_s_ was reduced by 22% on average at eCO_2_ without any significant change in stomatal density, indicating that the change in stomatal aperture rather than density determines the response of g_s_ to eCO_2_ [34]. Our study also confirmed that g_s_ was linearly decreased by eCO_2_ in both low N and high N treatments. Compared with low N, high N increased the slope and intercept of the relationship between g_s_ and CO_2_ concentration relationship, indicating that the responses of g_s_ of *S. superba* seedlings to eCO_2_ was modified by N fertilization as expected in hypothesis 2.

Photosynthetic stimulation was maintained despite stomatal conductance was decreased especially in high N treatments, which is consistent with the previous finding [35]. The first hypothesis that Pn would be reduced much more at 1000 μmol·mol^−1^ CO_2_ than at 700 μmol·mol^−1^ CO_2_ was well verified in high N treatments, while, in low N treatments, Pn would decrease when CO_2_ concentration exceeded 733 μmol·mol^−1^. Pn of *S. superba* showed downward trend in the whole measurement time, with September < August < July, which could be due to the change of environmental factors, such as the highest temperature in August and the lowest humidity in September in open top chambers (OTCs). Different from our findings that eCO_2_ increased Pn in most treatments, Pn of *S. superba* was decreased by eCO_2_ in both subtropical [20] and tropical sites [21]. The variation of Pn responses of the same species may be caused by the measurement conditions. In fact, in our experiment, Pn was measured in an OTC with the same CO_2_ concentration corresponding to the treatment, whereas, in the previous studies, Pn was measured in ambient air without considering experimentally set CO_2_ concentration [20,21]. When plants growing in eCO_2_ were measured in ambient air, the change of background CO_2_ concentration may lead to underestimation of Pn. In this study, Pn of low N treatments was linearly increased by eCO_2_ only in July, while Pn of high N treatments increased linearly throughout July, August, and September, indicating that N modified the response of Pn to eCO_2_, which confirmed our second hypothesis. This was different to the findings of the previous study [28]; that is, N fertilization did not affect the effects of eCO_2_ on photosynthesis of *S. superba* seedlings. This is due to N-limiting in our study site and not N-limiting in their study, which was clarified when discussing the different responses of leaf N of *S. superba* seedlings to eCO_2_ and N fertilization above. In August and September, Pn increased and then decreased with elevation of CO_2_ in low N treatments. Interestingly the maximum of Pn was both under 733 μmol·mol^−1^ CO_2_, although the regression relationship was different (Figure 3), which may indicate that 733 μmol·mol^−1^ CO_2_ was a certain threshold concentration for the growth of *S. superba* without N fertilization, while there was no threshold of CO_2_ concentration for the positive linear correlation between CO_2_ concentration and Pn when N was applied.

Although it is commonly reported that eCO_2_ increase the rate of photosynthesis, low N availability could limit the enhancement of biomass accumulation [36]. In this study site, the soil total N content was 0.5%, which was relatively low. It was confirmed in this study that low N availability limited biomass accumulation and even eCO_2_ reaching 1000 μmol·mol^−1^ could not increase each organ and total biomass. Compared with plants treated in low N, when plants exposed to eCO_2_, high N could alleviate low N availability and significantly increase total biomass. N fertilization also increased the biomass of root, stem and leaf when plants were exposed to 1000 μmol·mol^−1^ CO_2_. Other studies related to *S. superba* have also obtained similar results: eCO_2_ had no effect on biomass, N fertilization promoted the growth [21] and biomass accumulation, and eCO_2_ and N fertilization had a synergistic effect on biomass [37].

## 4. Materials and Methods

### 4.1. Experimental Site and Design

The experiments were carried out in open top chambers (OTCs) of 2 m in diameter and 2.2 m in height, which was used to study the effects of elevated O_3_ on plants in the past few years, and the details of the OTCs could be found in Chen et al. [38] and Yu et al. [39]. These field OTCs were set up in 2013, located in the Qianyanzhou ecological station of the Chinese Academy of Sciences (115°04′13″E, 26°44′48″N), with a subtropical monsoon climate. The mean annual temperature, mean annual precipitation, and mean relative humidity was 17.8 °C, 1471.2 mm, and 83%, respectively. Soil of the region, weathered from red sandstone and mudstone, is classified as Typic Dystrudepts Udepts Inceptisols using US soil taxonomy [40].

Four different CO_2_ treatments were set according to RCPs of IPCC, as ambient air (about 400 μmol·mol^−1^), 550 μmol·mol^−1^, 750 μmol·mol^−1^, and 1000 μmol·mol^−1^, with three respective OTCs for each, and in total 12 OTCs. The concentration of CO_2_ inside the OTCs was monitored with an CO_2_ analyzer (FGD2-C-CO_2_, Shenzhen Xin Hairui Science and Technology Development Co., Ltd., Shenzhen, China). The actual CO_2_ concentration was shown in Figure 4. In each OTC, there were two nitrogen fertilization treatments, no nitrogen fertilization (low N), and 10 g·m^−2^·yr^−1^ (high N). In total, there were eight treatments, and, for each treatment, there were five replicated seedlings and a total of ten seedling in each OTC. The mean light intensity in OTCs was 25,050 Lux, and the mean air temperature and humidity was 26.56 °C and 83.6% in OTCs, during the experiments.

### 4.2. Plant Growth

One-year-old seedlings of *S. superba* were transplanted to flower pots (diameter: 30 cm) containing local soil under ambient air condition in April 2018. The baseline soil nutrients were analyzed before the experiment, and the soils contained organic matter content of 8.63 g·kg^−1^, total nitrogen of 500 mg·kg^−1^, available *p* of 1.58 mg·kg^−1^, available K of 20.1 mg·kg^−1^, and soil pH was 4.70. In the end of April, the seedlings with similar height and basal stem diameter were selected and moved in to OTCs. And seven days later, the plants were fumigated with ambient or eCO_2_ air. In each OTC, five plants were selected to receive nitrogen fertilization, and, on June 15, July 20, August 24, and September 20, 2018, 150 mL of KNO_3_ solution (6.8 g·L^−1^) per pot was added. The same volume of tap water was applied to the control plants as low N treatment at each time. The seedlings were watered with tap water as needed during the experiment.

### 4.3. Sampling, Physiological and Biochemical Measurements

To investigate the responses of *S. superba* to eCO_2_ and N fertilization, two last years’ fully expanded leaves (third to fifth leaf position from the apex) of the main stem per seedling, and these two leaves were developed before treatments began. Two seedlings per treatment were randomly selected to measure gas exchange parameters. Gas exchange parameters were measured once a month from July to September. On 26 October 2018, about five last year’s fully expanded leaves (third to eighth leaf position from the apex) of the main stem per seedling and three seedlings per treatment in each OTC were randomly selected to determine plant nutrient. Finally, two seedlings without sampling leaves were harvested, and different tissues were dried at 70 °C to constant weight for dry biomass determination.

#### 4.3.1. Plant C, N, and Carbohydrates

Oven-dried samples of different tissues were grounded to powder through a 2-mm sieve for C, N, and carbohydrates analyses. Plant N was decided by an automated Kjeldahl apparatus (KD310, Opsis, Sweden). To extract soluble sugars, powdered leaf (0.5 g, dry weight (dw)) was added to 50 mL distilled water, and then high-pressure steamed for 2 h. Starch was extracted with 0.1 g (dw) powdered material added 10 mL distilled water and 1 mL hydrochloric acid (2:1), and then incubated at 100 °C in a water bath for 8 h. After chilling to room temperature, the mixture was adjusted to neutral pH with 40% NaOH solution. Both of the mixtures were filtrated and diluted to a constant volume at a room temperature. Carbohydrates were represented by water-soluble sugar and starch. Carbohydrates were determined by injection of 10-uL sample volume into a high-performance liquid chromatography system using a Sugar-Pak 1 chromatographic column and a refractive index detector (Waters HPLC 2695, Milford, MA, USA). The column temperature was 70 °C, and distilled water was used as mobile phase (flow rate 0.6 mL/min).

#### 4.3.2. Gas Exchange Measurement

Gas exchange was determined by a LI-6400 portable photosynthesis system (Li-Cor, Lincoln, NE, USA). The system controlled photosynthetically active radiation (PAR) at light-saturating 1000 μmol·m^−2^ s^−1^ using 6400-2B red/blue light-emitting diode (LED) light source. The block temperature was set to the ambient average (25–30 °C). Relative humidity was controlled at 50–65%. The parameters were measured in OTC with ambient or elevated CO_2_ between 9:00 a.m. and 11:30 a.m. The air temperature was 29.2, 32.6, and 31.1 °C, and relative humidity was 81.4%, 71.7%, and 66.2%, respectively, in July, August, and September in OTCs before gas-exchange data collection. The mean light intensity was 26533, 28333, and 30,033 Lux in July, August, and September in OTCs before data collection. The gas exchange parameters included photosynthate rate (Pn, μmol·m^−2^ s^−1^), stomatal conductance (g_s_, μmol·m^−2^ s^−1^).

### 4.4. Statistics

CO_2_ treatment means were statistically compared, respectively, at low N and high N using the statistical package SPSS (SPSS Inc., Chicago, IL, USA) for leaf nutrition. The relationship between CO_2_ concentration and means of leaf N, starch, soluble sugar, and C:N were analyzed using Pearson correlation, respectively, at low N and high N. Pearson correlation was also used to determine the relationship between CO_2_ concentration and Pn and g_s_. If there was significant correlation, the linear equation was generated by OriginPro 9.0. While, for Pn at low N in August and September, quadratic regression equation was used to explain the relationship between Pn and CO_2_ concentration, biomass was analyzed by using one-way ANOVA, Tukey’s HSD to determine the effects of CO_2_ treatments, respectively, at low N and high N, and the effects of N fertilization were determined by *t*-test at each CO_2_ level. The single and internal effects of eCO_2_ and N fertilization on leaf nutrition and biomass, and eCO_2_, N fertilization, and measuring date on Pn and g_s_ were determined by a multi-way ANOVA. All the figures were produced by OriginPro 9.0.

## 5. Conclusions

The responses of *S. superba* to eCO_2_ are mediated by N fertilization. Although N fertilization increased leaf N of *S. superba*, leaf N was decreased by eCO_2_ in both low N and high N, and N fertilization accelerated this reduction with larger slope of eCO_2_ and leaf N relationship. High N decreased leaf C:N compared to low N, and the slope and intercept of leaf C:N and eCO_2_ relationship were both decreased, indicating N fertilization mitigated the negative effects of eCO_2_ on the imbalance of leaf C and N. N fertilization also modified the responses of Pn and g_s_ to eCO_2_. In low N treatments, Pn was improved with CO_2_ concentration increasing in July, while, in August and September, Pn was increased and then decreased by eCO_2_ with a threshold of 733 μmol·mol^−1^ CO_2_, and, in high N treatments, Pn was consistently increased by eCO_2_ in all of July, August, and September. Through these ways, N fertilization application in this low N availability site significantly accumulated the biomass of the *S. superba* seedlings. Although this present study lasted only one growth season, the results indicated that global change with multi-factors, like both eCO_2_ and N deposition, could have stronger effects on growth of woody plants, which should be paid continuous attention.

## Figures and Tables

**Figure 1 plants-10-00383-f001:**
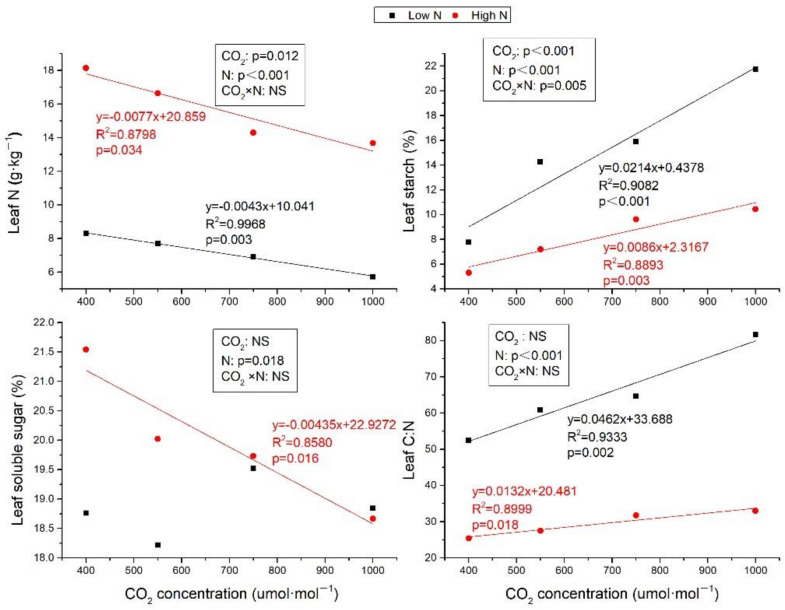
The different effects of elevated CO_2_ on leaf N, carbohydrates and leaf C:N between low N and high N treatments. The results of two-factor analysis were also shown. NS: not significant. Leaf N, soluble sugar, and starch were all calculated by dry mass. Values are means ± SE (n = 6).

**Figure 2 plants-10-00383-f002:**
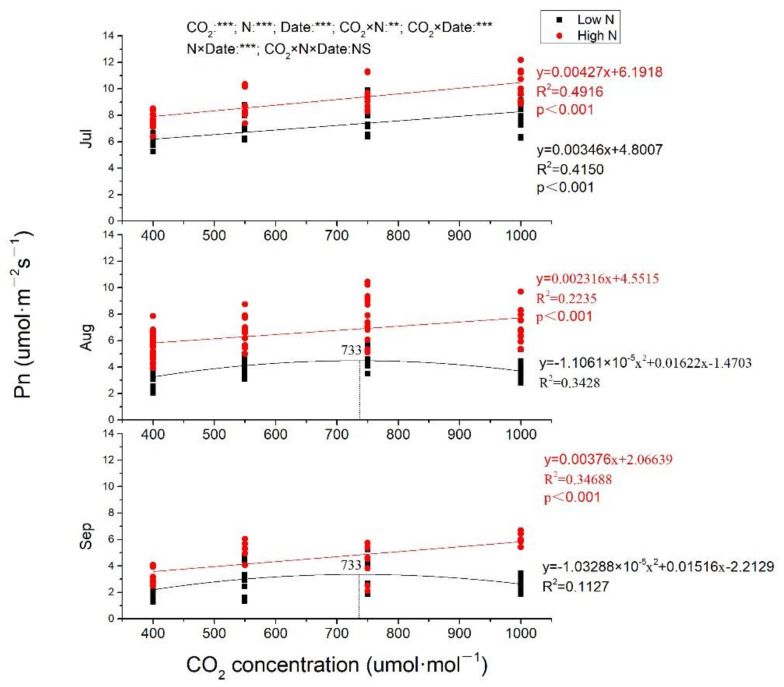
The different responses of photosynthate rate to elevated CO_2_ at low N and high N treatments. Pn, photosynthate rate. CO_2_: different CO_2_ concentrations; N: N fertilization; Date: measuring time; ***, **: significant at 0.001 and 0.01 level; NS: not significant; Jul: July; Aug: August; Sep: September.

**Figure 3 plants-10-00383-f003:**
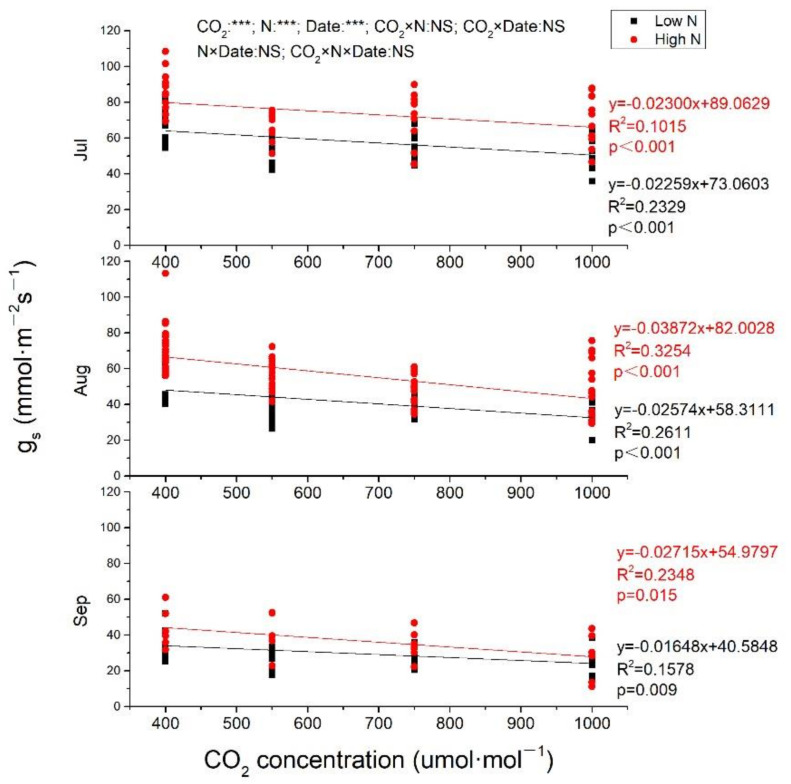
The effects of elevated CO_2_ and N fertilization on stomatal conductance. g_s_, stomatal conductance. CO_2_: different CO_2_ concentrations; N: N fertilization; Date: measuring date; ***: significant at 0.001 level; NS: not significant. Jul: July; Aug: August; Sep: September.

**Figure 4 plants-10-00383-f004:**
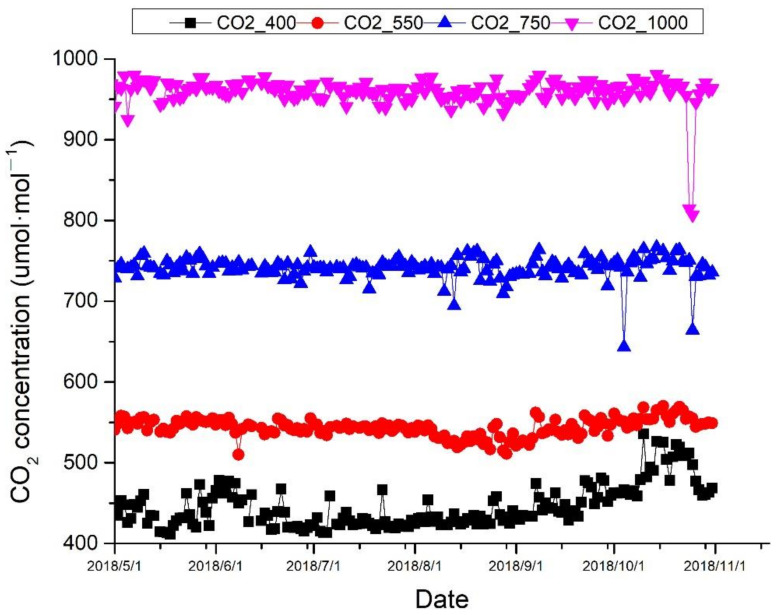
The actual CO_2_ concentration at different treatments during the growth season.

**Table 1 plants-10-00383-t001:** Effects of elevated CO_2_ and N fertilization on plant biomass. *p*-Values (in the brackets) indicate significant differences between mean values of low N and high N at the same CO_2_ concentration. For the main effect of either CO2 or N fertilization, and the interaction between CO_2_ and N fertilization, *p*-values indicate significant effects of that factor on these biomass variables; “NS” indicate no significant effects of a factor on biomass. dw, dry weight; Root, root biomass; Stem, stem biomass; Leaf, leaf biomass; Total, total biomass. CO_2_, different CO_2_ concentration treatments; N, different N fertilization treatments; CO_2_ × N, the interaction of CO_2_ concentration and N fertilization.

CO_2_ (μmol·mol^−1^)	Biomass (g dw)				
		Root	Stem	Leaf	Total
ambient	Low N	6.77 ± 0.89	5.37 ± 0.87	6.17 ± 1.08	18.30 ±2.73
	High N	7.55 ± 0.66 (0.519)	8.63 ± 1.52 (0.135)	9.55 ± 1.52 (0.144)	25.73 ±3.60 (0.176)
550	Low N	5.50 ± 1.05	5.18 ± 0.45	5.30 ± 1.15	15.98 ±2.60
	High N	9.38 ± 1.75 (0.130)	7.10 ± 0.40 (0.063)	10.13 ± 0.52 (0.019)	26.62 ± 2.83 (0.05)
750	Low N	5.42 ± 0.83	5.85 ± 0.81	5.53 ± 0.49	16.80 ± 1.35
	High N	11.70 ± 1.70 (0.30)	12.42 ± 2.65 (0.077)	12.17 ± 2.44 (0.056)	36.28 ± 6.51 (0.043)
1000	Low N	6.98 ± 1.02	5.95 ± 1.2	6.12 ± 0.91	19.05 ± 2.96
	High N	13.60 ± 1.94 (0.039)	12.45 ± 0.60 (0.008)	13.27 ± 0.15 (0.014)	39.32 ± 1.32 (0.003)
CO_2_		NS	NS	NS	NS
N		<0.001	<0.001	<0.0001	<0.0001
CO_2_ × N		NS	NS	NS	NS

## Data Availability

The data presented in this study is available on request from the corresponding author.

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
