# Peer review of "Nitrogen Fertilization Modified the Responses of Schima superba Seedlings to Elevated CO2 in Subtropical China"

_plants, 2021, doi:10.3390/plants10020383_

Round 1

Reviewer 1 Report

I propose to shorten the title of the article "The growth of Schima superba seedlings responded significantly to elevated CO2 in combination with nitrogen fertilization" is sufficient. Why specify a non-significant factor in the title? In my opinion in Conclusions section should be more obtained results.

Author Response

Response: The title is revised to “Nitrogen fertilization modified the responses of Schima superba seedlings to elevated CO2 in subtropical China”. The conclusion section is also revised with more results as suggested.

Reviewer 2 Report

Dear authors,

the manuscript isinteresting and the experiment has been quite well structured; the object of study is very attractive, and it is in accord to the increasing interest in climate change; all sections of the paper are quite well described, except the results. In fact, although the results are interesting, their graphical representation is incorrect and little clear. In particular, the figures must be according to the results of statistical analysis, reporting the interaction of third or second degree only if significant. In addition, the graphical representation of the data (line graph) is incorrect, because the authors don’t report any trends; I would suggest to change the graph (histograms) or report the data in tables. Therefore the manuscript can be accepted after major revisions.

Author Response

Response: Thanks very much for the kind comments. The figures are remade and tables were deleted. In order to clearly show the different responses of plants to eCO2 (including leaf N, starch, leaf C/N, Pn and gs, Figure 1-3) at low N and high N, we remade the figures with accompanying regression equations separated for low vs. high N. And the bar graph was used to present the effects of eCO2 on biomass (Figure 4) with different colored bars for low and high N.

Reviewer 3 Report

See comments in attached file.

Round 2

Reviewer 2 Report

Dear authors,

the paper has been notably improved however, there are few observations.

I would suggest using the same scale for the y-axis both in figures 2 and 3.

Figure 4: the figure is incorrect because the interaction N x CO2 isn't significant (see line 127). The authors should represent the data differently, for example, the main effects. 

Line 177: is there a mistake? 733 and 700?

Line 253: "photosynthetic pigments" are a mistake? the authors don't report these data

Author Response

I would suggest using the same scale for the y-axis both in figures 2 and 3.

Response:Thanks for the suggestion. The scale for the y-axis in both figure 2 and 3 are unified. 

Figure 4: the figure is incorrect because the interaction N x CO2 isn't significant (see line 127). The authors should represent the data differently, for example, the main effects. 

Response: Sorry for the mistake. We re-analyzed the data and used t-test to determine the effects of N fertilization. The results were represented in Table 1. Accordingly, we rephrased some description about the effects of biomass, like in line 19, 124-129, 216-219 and line 323.

Line 177: is there a mistake? 733 and 700?

Response: Sorry for the mistake, 700 was deleted in line 186 in revised paper.

Line 253: "photosynthetic pigments" are a mistake? the authors don't report these data

Response: "photosynthetic pigments" was deleted.